# Quasi-Distributed Fiber Sensor-Based Approach for Pipeline Health Monitoring: Generating and Analyzing Physics-Based Simulation Datasets for Classification

**DOI:** 10.3390/s23125410

**Published:** 2023-06-07

**Authors:** Pengdi Zhang, Abhishek Venketeswaran, Ruishu F. Wright, Nageswara Lalam, Enrico Sarcinelli, Paul R. Ohodnicki

**Affiliations:** 1Mechanical Engineering and Materials Science, University of Pittsburgh, 3700 O’Hara Street, Pittsburgh, PA 15261, USA; pez37@pitt.edu (P.Z.); abv20@pitt.edu (A.V.); ens65@pitt.edu (E.S.); 2National Energy Technology Laboratory, 626 Cochrans Mill Road, Pittsburgh, PA 15236, USA; ruishu.wright@netl.doe.gov (R.F.W.); negeswara.lalam@netl.doe.gov (N.L.); 3Electrical and Computer Engineering, University of Pittsburgh, 3700 O’Hara Street, Pittsburgh, PA 15261, USA

**Keywords:** distributed optical fiber sensing system, defect detection, pipelines, physics-informed datasets, simulations, welding detection, sensing system, data classification performance and noise robustness

## Abstract

This study presents a framework for detecting mechanical damage in pipelines, focusing on generating simulated data and sampling to emulate distributed acoustic sensing (DAS) system responses. The workflow transforms simulated ultrasonic guided wave (UGW) responses into DAS or quasi-DAS system responses to create a physically robust dataset for pipeline event classification, including welds, clips, and corrosion defects. This investigation examines the effects of sensing systems and noise on classification performance, emphasizing the importance of selecting the appropriate sensing system for a specific application. The framework shows the robustness of different sensor number deployments to experimentally relevant noise levels, demonstrating its applicability in real-world scenarios where noise is present. Overall, this study contributes to the development of a more reliable and effective method for detecting mechanical damage to pipelines by emphasizing the generation and utilization of simulated DAS system responses for pipeline classification efforts. The results on the effects of sensing systems and noise on classification performance further enhance the robustness and reliability of the framework.

## 1. Introduction

Pipelines play a vital role in transporting and distributing liquid and gaseous fuels across industries such as oil, gas, and petrochemicals. As they traverse diverse and challenging terrains, maintaining their structural integrity becomes a daunting task. To ensure national security and economic growth, it is imperative to develop advanced monitoring methods to counter external threats, such as sabotage, unauthorized access, construction accidents, and natural disasters, as well as internal degradation due to factors such as corrosion, erosion, and fatigue. Structural health monitoring (SHM) is an innovative technology that blends sophisticated sensor systems with intelligent algorithms to assess a structure’s “health”, thereby increasing reliability, safety, and automation capabilities while lowering lifecycle costs. Consequently, SHM has garnered significant attention as a promising solution for improving structural integrity in civil infrastructure, aerospace, and mechanical systems. One particularly compelling application is damage detection through guided wave nondestructive testing, where ultrasonic sensors identify structural damage or changes, such as clamp additions or weld presence, by detecting backscattering acoustic responses.

Elastic perturbations known as guided waves are capable of propagating over extended distances in thin-walled structures while experiencing minimal amplitude loss. Laboratory experiments [1] have demonstrated the efficiency of using guided acoustic waves to detect and locate pipeline anomalies in critical areas that are susceptible to defects. To excite a cylindrical structure and propagate UGWs, an appropriate signal is chosen, which will then propagate through the structure and encounter any damage or other material discontinuities. Traditionally, by analyzing the resulting changes in the waveform, the location and severity of the damage can be quantified. It is important to note that UGWs can experience dispersion as they propagate, which can affect the accuracy of damage detection. To analyze the dispersion of UGWs, a MATLAB toolbox package named PCDISP [2] is commonly used. This toolbox utilizes the Navier–Lamé equation [3], which describes the behavior of elastic waves in solid bodies, to calculate the dispersion curves of different guided wave modes. The use of PCDISP allows for a more accurate analysis of the dispersion and ultimately a more accurate identification of the location and severity of the damage. By utilizing the appropriate excitation signal, analyzing the resulting changes in the waveform, and utilizing tools such as PCDISP, it is possible to effectively detect and quantify damage in the structure. Wave phase velocities and group velocities of a steel pipeline with an outside diameter of 12 inches and wall thickness of 0.5 inches are shown in Figure 1a,b.

Moreover, guided wave NDE has the potential to significantly decrease the number of sensors necessary for monitoring a structure. In guided acoustic wave sensing, the exciting transducer is therefore typically also used as the measurement sensor, thereby measuring the backscattered acoustic wave. Such an installation scenario can be highly limiting in terms of investigating damage over large distances and in remote locations, and the amount of information that can be extracted is also limited by what can be measured at the excitation location. The use of distributed fiber optic sensors as measurement transducers has been proposed as an alternative solution, and they can be placed at multiple points along a pipeline to monitor for damage. Practical limitations of permanently mounting conventional NDE sensors on structures have been widely discussed [4]. Assuming sufficiently high-frequency bandwidth can be achieved, distributed fiber optic sensors are promising candidates for guided acoustic wave detection schemes because they are able to detect acoustic signatures generated by external events or alternatively scattered by defects within a pipeline segment with high spatial and temporal resolution [5,6,7]. In the past decades, distributed fiber vibration/acoustic sensor technology has gained increasing attention and tremendous growth. The quasi-distributed/point fiber vibration/acoustic sensor technologies include fiber Bragg grating (FBG) [8], Fabry–Pérot [9], and multimodal interference [10]. Various interferometric configurations include the Sagnac interferometer [11], the Mach–Zehnder interferometer (MZI) [12], and polarization-OTDR (POTDR) [13]. Fiber optic sensors are known for being resilient in harsh environments and capable of distributed interrogation. They have a unique ability to perform under distributed strain, and acoustic measurements using backscattered light phenomena in unmodified telecommunications fibers or fibers that have been modified to enhance the scattering can result in high sensing performance with improved spatial resolution measurements [14,15]. The most common distributed acoustic sensing modality involves a technique referred to as phase–optical time domain reflectometry (φ—OTDR), as discussed in our recent review of distributed optical fiber sensing [16]. The Φ-OTDR system utilizes the interference effects within pump pulses generated by a narrow linewidth (usually <10 kHz) laser source. The Φ-OTDR system demodulates the backscattered Rayleigh signal amplitude/phase to acquire acoustic signals. The Rayleigh signal amplitude varies with the strain on the sensing fiber induced by the surrounding acoustic signals. In the use of a fiber optic-based distributed acoustic sensing (DAS) system, the fiber optic cable is used as the sensor in which a pulsed laser is used to excite the fiber and backscattered light is detected and processed using appropriate optical interrogator hardware.

Machine learning (ML) techniques can be instrumental in processing raw data acquired from fiber acoustic sensing, providing benefits in classification, pattern recognition, prediction, and system optimization. These are especially useful when the relationship between inputs and outputs is not mathematically explicit [17]. Numerous studies have focused on improving the dynamic range, spatial resolution, and sensitivity of Distributed Acoustic Sensing (DAS) system hardware and associated signal processing methods. The integration of DAS with Pattern Recognition Systems (PRS), semi-supervised k-means clustering for structural integrity assessment, and advanced event recognition methodologies, such as Convolutional Neural Networks (CNNs) and deep learning, have emerged as areas of particular interest [18,19,20,21]. The implementation of the Squeeze and Excitation WaveNet (SE-WaveNet) model in threat identification further underscores the potential of these techniques for real-time surveillance [22]. However, the effectiveness of these ML techniques heavily relies on the quality and abundance of their training datasets, as well as the precision of sensor signals [17]. A crucial hurdle in the widespread adoption of ML-based data analysis for Structural Health Monitoring (SHM) is the costliness and time-consuming nature of acquiring training datasets of damage/failure events for real-world structural systems. Although data for commonplace events such as human or vehicle activity can be easily collected, real-time sensor data for pipeline structural damage or failure events proves to be exceptionally challenging. Furthermore, the diversity of potential damage events can complicate ML model training and hinder performance.

This challenge can be addressed by training ML models on simulated sensor datasets, which integrate a sensor measurement model (e.g., distributed acoustic sensor measurement model) and a high-fidelity physics-based numerical model simulation of the structural degradation event (e.g., ultrasonic vibrations of a pipeline with corrosion damage). This strategy allows for systematic simulation of sensor datasets for various pipeline degradation events, upon which ML classifiers can be trained for predicting structural degradation events of interest. The performance of these ML classifiers can further be enhanced by supplementing training data with available experimental measurements, a concept known as domain adaptation in ML literature, which is gaining traction in SHM [23]. In the case of optical fiber sensing systems in particular, a need exists for simulated DAS/Quasi-DAS system signal generation to enhance the data available and boost pipeline health monitoring capability, facilitating the creation of comprehensive, high-quality datasets for training and improving accuracy for real-world applications.

In this paper, we propose a unique simulation approach aimed at bridging Guided Wave simulation and the simulation outcomes from DAS/Quasi-DAS systems, and we employ a classification method that utilizes a time–space data matrix for a fully distributed sensor system. The focus of this work is on generating simulation datasets, an approach that has shown significant potential in applications such as pipeline structural health monitoring (SHM) and defect identification using Distributed Acoustic Sensing (DAS) data. The application of physics-based simulation datasets coupled with DAS systems can notably improve the accuracy of SHM by enabling real-time data analysis to detect and identify defects in pipelines. Differing from traditional methods, the generation of physics-based simulation datasets allows for extraction and identification of patterns within raw data that may be challenging to detect through conventional means. This feature proves especially valuable when dealing with large volumes of data generated by DAS systems in real-time. SHM systems that utilize physics-based simulation datasets are also capable of adapting to variable conditions, making them well-suited to complex environments where traditional strategies may fall short. In essence, integration of physics-based simulation datasets with DAS systems holds the potential to significantly enhance pipeline SHM and defect identification by enabling accurate and timely analysis of extensive sensor data [17].

## 2. Physics-Based Modeling Enhancing Guided Wave Approaches

Our current study endeavors to demonstrate the value of incorporating data-driven techniques into guided wave approaches for damage detection, leveraging physics-informed data sets relevant to distributed fiber optic sensing. This is depicted in Figure 2, where our learning framework integrates the following steps:(a)The generation of simulated data sets representative of various mechanical damages/defects, incorporating multi-physics constraints such as conservation laws, boundary conditions, and pipeline types.(b)The simulation of Guided Wave Ultrasonic (GWU) responses and Distributed Acoustic Sensing (DAS) responses in the grey box. This step is crucial as it bridges physics-based modeling and machine learning modeling. Simulated DAS responses, in particular, provide a unique opportunity to train our machine learning model with a rich, nuanced dataset that mimics real-world pipeline defect scenarios.(c)Pre-processing of the simulated data, which includes noise consideration to account for real-world variabilities and uncertainties.(d)Training of the machine learning model, using the pre-processed simulated DAS response data. This step is essential for knowledge discovery associated with mechanical damage detection and identification.(e)Application of the trained model to test data for damage classification, demonstrating the potential of our approach for practical applications.

In our future work, we plan to augment our framework with designed experiments, comparing their results against our simulation predictions. This will allow us to further validate and refine our proposed methodologies and to integrate experimental results into our approach. This ensures a more comprehensive, synergistic method of damage detection and identification in pipeline systems.

**Figure 2 sensors-23-05410-f002:**
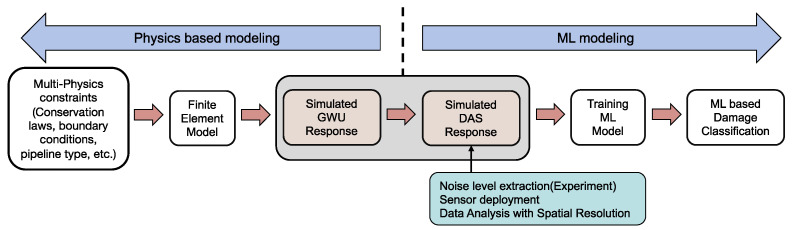
The framework for the physics-based modeling method of structure change detection consists of the above components.

## 3. Generating Datasets Using Guided Ultrasonic Wave (UGW) Approaches

### 3.1. The Excitation Concept of Guided Ultrasonic Waves (UGWs)

Structural health monitoring (SHM) is a crucial approach to detecting and quantifying damage in structures. One of the effective ways to accomplish this is by using UGWs, a non-destructive testing technique that is based on the analysis of changes in the waveform of an excited signal. A guided wave propagates along a hollow cylindrical shell in a pattern, as illustrated in Figure 2a,b, while different damage location and severity could cause wave scattering in the form of mode conversion as well as reflection and transmission. Cylindrical structures are particularly suitable for this approach because they can support different types of UGWs, such as axisymmetric and longitudinal (L mode), axisymmetric and torsional (T mode), and non-axisymmetric and flexural modes (F mode) [3]. As a result, linking changes of wave modes with associated damage types, localization, and severity makes them flexible for damage identification. Much research [24,25] has demonstrated that certain guided wave modes are highly sensitive to minor damage, including damage difficult to detect by other non-destructive detection methods. In addition, guided waves are used for damage identification, as an elastic vibration in thin plate-like structures, due to low attenuation rate, high penetration capability, ease of generation, and ease of use, are also highly sensitive to damage in small-sized structures [25,26]. Since guided waves generate stresses over the entire thickness of the plate, it is possible to interrogate the entire plate thickness. This means that defects starting at the surface of the plate as well as internal defects can be detected. As guided waves can have more than one propagation mode, and even when a single mode guided wave interacts with a structural defect, the received signal usually contains more than one mode, at which point the proportion of different modes present in the wave depends on the mode transition and other impedance changes at the defect which impact received signals [27].

To excite a cylindrical structure and propagate UGWs, an appropriate signal is chosen, which will then propagate through the structure and encounter any damage or other material discontinuities. It is important to note that UGWs can experience dispersion as they propagate, which can affect the accuracy of damage detection.

The L (0, 2) mode is selected for excitation, due to its high sensitivity to circumferential cracks and low identification difficulties [28]. This mode is known to have a faster velocity than L (0, 1) at 50 kHz and is reported to be more sensitive to the circumferential size of pipeline defects. The selected frequency for the L (0, 2) mode allows for flat dispersion curves as illustrated in Figure 2, minimizing dispersion during propagation. The excitation signal used for this purpose is a 50 kHz, 5 cycle sinusoidal signal modulated with a Hanning window in the axial direction, with an amplitude of 0.003 inches, representative of a typical excitation achievable with a guided wave collar:(1)$$u\left(t\right)={\bar{u}}_{amp}\left(1-cos\frac{2\pi f_{c}t}{n}\right)\sin\left(2\pi f_{c}t\right);$$
where ${\bar{u}}_{amp}$ is amplitude of the signal, $f_{c}$ is frequency. Specifically, in this equation provided, n is the number of cycles of the signal that should be included in the excitation (based upon the Hanning window (period)). The assumed excitation amplitude is 0.003 inches, based on calibration between piezo actuator voltage and simulation excitation displacement from past work [28,29]. This signal was defined in ANSYS to simulate the effects of an actuator, which is a representative excitation waveform when compared with existing commercial ultrasonic acoustic NDE technologies [28,30]. The excitation signals in time and frequency domain are shown in Figure 3a,b, respectively.

### 3.2. Finite Element Modeling and Wave Propagation Analysis of Steel Pipe Structure

To construct the finite element model of the steel pipe structure, the dimensions and material properties of the pipe and excitation source are established, and then calibrated based on previous research by P.S. Lowe, [31]. Table 1 and Table 2 show the dimensional and material parameters of the pipe structure that was tested. Figure 4a shows the overall schematic setup of the finite element simulation, as well as an example of the visualization of the finite element simulation of the investigated pipe. In addition, the figure shows that the guided wave propagates along the pipe from the left side to the right and produces dispersion phenomena. When utilizing NDT technology that combines ultrasonic wave guides with fiber optic sensors, the information is transmitted to the signal receiver by the sensor while the excitation signal is passing through.

The model featured a 96-inch-long nominal 96-inch schedule 40 steel pipe, with an outer diameter of 12 inches and a wall thickness of 0.5 inches. The assumed material properties for steel are presented in Table 2. In the current simulation setup, our study focuses on a single material steel pipeline structure, and we assume negligible damping for simplicity due to strong dependence upon the details of the experimental configuration. For boundary conditions, both ends of the pipeline were assumed to be fixed, leading to zero displacements and rotation at those points. To simulate sensor data, the displacement component of the pipeline was extracted from the numerical simulations. This information was then used to recreate sensor readings along the pipeline, providing valuable insights into the pipeline’s response to various types of excitations and damage scenarios.

An averaging scheme was employed to generate sensor data from the displacement values obtained from the simulation. In this method, the sensor signal is calculated as an average over specific segments to ensure a more accurate representation of the sensor readings, consistent with a first-order approximation to the measured sensor response of an optical fiber distributed sensor “gauge length” or a quasi-distributed sensor active sensing length. This averaging process takes into consideration potential variations in the data due to noise, sensor positioning, and other factors. A detailed discussion of this approach can be found in Section 4. In the current analysis, guided wave damping attenuation is not considered due to the complexities involved in modeling, with the focus being on the proof of concept and feasibility of the modeling approach. However, the effectiveness of structural detection will ultimately rely on the damping characteristics and their spectral response to target damage. Future investigations will incorporate damping based on comparisons with experiments, specifically tailored to scenarios related to pipe installation (e.g., buried in soil, secured with clamps, etc.), as elaborated in Section 5.

To explore the characteristics of waves generated by the assumed excitation signal and to understand wave propagation, a 3D model was constructed using ANSYS Finite Element (FE) software. For the numerical simulation of guided wave-based pipe structures, the selection of analysis steps and time increments relies on the system properties (including dimensions and excitation signal), as well as the sampling interval and frequency of detection signals. The Transient Structural Module, which is a dynamic implicit analysis step in ANSYS, was chosen to examine the problem of integrating damage identification for pipe structures with fiber optic sensor deployment [6]. The guided wave propagation can be derived from the Navier-Lame equation [3]:(2)$$\mu \nabla^{2}u+\left(\lambda +\mu \right)\nabla \left(\nabla \cdot u\right)=\rho \left(\frac{\partial^{2}u}{\partial t^{2}}\right),$$
where u is the displacement, t is the time, ρ is density, $\nabla^{2}$ is the 3-dimensional Laplace operator, λ and μ are Lamé’s constant. Guided wave generation and propagation in a cylindrical structure can be simulated using numerical methods. The equation of motion can be expressed in matrix form:(3)$$M\ddot{u}+C\dot{u}+Ku=F,$$

In Equation (3), *M* represents the structural mass, *C* represents the damping, and *K* represents the stiffness, all in matrix form. *F* is a vector of excitation or loaded force. It is worth noting that $\dot{u}$ and $\ddot{u}$ are the first- and second-time derivatives of displacement, respectively. In our case, the damping factor is not considered. Equation (3) can be solved by the Newmark time increment method. The time step, Δt, is the step size in Equation (3), and the smaller the time step, the higher the accuracy of the model. For a trade-off between accuracy and calculation time [32], the rule of time step is expressed:(4)$$\Delta t<\frac{1}{20f_{max}},$$

Here, Δ*t* denotes the time increment and $f_{max}$ represent the highest frequency of the excitation signal. So, the maximum time increment should be smaller than 1/20 of the excitation signal period corresponding to the highest frequency. In this paper, the highest frequency of the excitation signal is 50 kHz, so it can be obtained that the time increment should be equal or less than 1 µs by Equation (4). Considering the accuracy of the analytical results, 1 µs is chosen as the increment.

To ensure the accuracy of the calculation and consider the influence of calculation efficiency, in this paper, the general mesh size should be smaller than 1/20 [28] of the minimum wavelength along the pipeline body, as shown in Equation (5)
(5)$$l_{m}<\frac{\lambda_{min}}{20},$$
where $l_{m}$ means the size of the mesh elements and $\lambda_{min}$ denotes the minimum wavelength of the guided wave. Considering Equation (5) and considering the structure of pipeline, this requires that the mesh size should be smaller than 3.93 inches and the thickness of pipeline should include a minimum of between 2 and 3 elements. Therefore, the meshing size of the pipeline is set to 0.2 inches and the meshing method is set to sweeping with edge division to ensure the pipe mesh has three layers of mesh in the thickness direction. In addition to the general setting for healthy structures, the element size for defect regions should be less than one third of minimum defect geometry parameter.

### 3.3. Wave Propagation Analysis of Steel Pipe Structure

Finite element analysis has been performed to investigate the nature of the excited modes from the assumed excitation signal. Given the frequency range of excitation and the assumed longitudinal displacement direction of the excitation signal as well as the symmetric nature of the signal around the pipe circumference, the L (0, 1) and L (0, 2) modes are anticipated to be the dominant excited modes. Prior literature [28,31] describes the potential of isolated excitation of L modes as preferred for defect identification with conventional ultrasonic guided NDE application. Longitudinal mode guided waves were reported to be sensitive to circumferential dimensions of pipe defects. When the circumferential length of the defect increases, all else being the same, the signal reflection area increases, which enhances the reflected signal. The L (0, 2) wave mode shows predominantly axial displacement, and the L (0, 1) wave mode exhibits a dominant radial displacement with lower axial displacement [33].

Commercially available UGW transducers (axially aligned) can therefore be used to excite the L (0, 2) based upon prior work [31]. Predicted waveforms are labelled based on the time-of-flight information extracted from dispersion curves shown in Figure 2. The FE model was performed to study the waveform generated by excitation consistent with current commercially available UGW transducers by applying the vibration longitudinally as in Figure 4a. Figure 4b shows the predicted time-domain data from FE and the displacement caused by the axial excitation. To validate the simulation setup described in the study, the authors reference the work of P.S. Lowe. Lowe conducted experiments on an 8-inch schedule 40 steel pipe, with an outer diameter of 12 inches and a wall thickness of 0.5 inches. A ten-cycle Hanning-windowed 40 kHz pulse was used as the excitation signal, and the generated waveform was captured 1 m away from the point of excitation. To compare the results of their simulation with these experimental results, the authors used the same simulation setup and assumed excitation signal. Results of the simulation are shown in Figure 4c and demonstrate good agreement with experiment, as indicated by the close match between predicted results from the finite element (FE) simulation (represented by the blue line) and experimental results (represented by the red line) in [25].

## 4. Sensor Measurement Model

A quasi-distributed acoustic sensing system uses a fiber optic cable as a sensor to measure local dynamic straining along the length of the fiber. A pulsed laser excites the fiber, and backscattered light is detected and processed using an optical interrogator, typically using phase-optical time domain reflectometry (φ—OTDR). The smallest resolvable spatial resolution of the optical fiber segment is referred to as the “gauge length”, which can be adjusted to optimize the signal-to-noise ratio (SNR) and frequency resolution. Unlike fully distributed acoustic sensing systems, quasi-distributed systems are not able to resolve acoustic signals along the entire length of the fiber. In prior studies, the ratio of gauge length and spatial wavelength of the acoustic excitation signal for fully distributed acoustic sensors was used to achieve a trade-off between SNR and frequency distortion [5].

When we consider fiber optic-based distributed acoustic sensing system in our simulation work, the concept of gauge length needs to be incorporated into the development of the sensor measurement model. In optical fiber, fully distributed optical sensing involves monitoring the intensity and phase of the backscattered light and analyzing as a function of time to allow measurement of local dynamic strain along the length of the fiber. The minimum resolvable spatial resolution of the fiber segment is dictated by the physical hardware limitations as well as the signal processing scheme applied and is referred to as the “gauge length”. The measurement length of a given distributed fiber optic sensor can be adjusted over reasonable limits, and the selection of an optimized gauge length is an important aspect of data acquisition and processing. For example, gauge length has a significant effect on the signal-to-noise ratio (SNR) of the data and on the accurate resolution of an acoustic wave within the frequency domain. Dean et al. [34] presented an approach which discussed an explicit trade-off between the maximum SNR and the estimated measurement wavelength by setting an optimum gauge length $GL_{opt}$:(6)$$GL_{opt}=\frac{R\upsilon }{f_{p}}$$
where *R* denotes ratio of gauge length and spatial wavelength of acoustic excitation signal, 𝑣 donates acoustic velocity (m/s), and 𝑓_𝑝_ is peak frequency (Hz); In our simulation, the resulting signal is extracted using an assumed gauge length of 2.21 inches with *R* equal to 0.49 and a target wave mode velocity of 5161 m/s to achieve a trade-off between signal-noise ratio and frequency distortion, see detail in T. Dean and T. Cuny (2017) [34].

Based on the resolution limit we cited and the assumed gauge length in our simulation, the resulting signal can be treated as an averaged strain value over a certain fiber segment. Specifically, the averaging strain value can be expressed as the integral of the local dynamic strain over the segment length L, divided by the segment length L. This averaging operation is necessary to ensure that the signal is representative of the entire segment, and to minimize the effects of noise and frequency distortion:(7)$$\;\epsilon_{avg}=\frac{1}{L}\int_{x}^{x+L}\epsilon \left(x\right)dx$$

Here, $\epsilon_{avg}$ is the averaged strain value, *x* is the spatial coordinate along the fiber, and $\epsilon \left(x\right)$ is the local dynamic strain at the point *x*. Additionally, typical gauge lengths of fully distributed interrogation systems and limitations in upper frequency bandwidth are not optimized for features we are trying to measure, due to short wavelength and high frequency of UGW (ultrasonic guided waves). Hence, our simulations assume a quasi-distributed sensing scheme based on fiber Bragg gratings, in-line Fabry–Perot interferometers, or other interferometric structures, which can achieve smaller gauge lengths, better optimization for measuring UGW features, and higher bandwidth acoustic frequency sensing as required for UGW monitoring [5,34].

## 5. Data Modeling

### 5.1. Simulation Collection for Pipeline Structure Changes

Structural discontinuities can arise from variations in material properties, such as a structure that is partially embedded in a surrounding medium. To represent practical scenarios, this study considers three types of pipeline events: welds, clamps, and corrosion defects. The weld is modeled as a narrow cylinder with a constant inside diameter that protrudes through the weld and connects to the pipe, with an outside diameter larger than that of the pipe. The clamps on the pipeline are modeled with a specific surface connected to the pipe and a stiffness ratio determined by constraint of the clamps. The categorization of corrosion used in the work is based on the classification proposed by M. Askar [35], which includes three main types: localized, general, and pitting corrosion. These corrosion types will be described in more detail in the following sections. As Figure 5 shows, localized corrosion is mainly due to damage to the surface in the form of mass removal in selected areas, resulting in formation of pits, cracks, and grooves. Pitting is a form of localized corrosion damage that results in the formation of small defects or pits. We differentiate between types of corrosion due to their significant differences in size. Pitting corrosion typically has a size in the hundreds of micrometers range, making it a challenge for finite element analysis (FEA) models due to the need for fine meshing in proximity to the defect and relatively weak scattering signature. General corrosion is another type that occurs in a relatively large area and is caused by several electrochemical processes occurring consistently over the entire surface under consideration. In this type, key characteristics are the loss of metal thickness and unit weight, both of which can have a measurable signature in an acoustic signal to reflect specific characteristics.

By conducting multiple simulation runs for various types of defects in pipelines across five groups—Clamp, Welding, Localized Corrosion, General Corrosion, and Pitting –valuable data is gathered for each group, with varying variables specified in Table 3.
Clamp: Elastic support loaded by clamps; varying axial length (0.5~5 mm) and stiffness factor ($5\times 10^{6}$~$6.5\times 10^{6}$ N/m).Welding: Discontinuity and material property changes between pipeline and welded portion; varying axial length (0.5~5 mm), height (0.5~5 mm).Localized Corrosion: Presence of a rectangular notch on the pipeline’s inner surface; varying axial length (25.4~152 mm) and depth (0.5~5 mm).General Corrosion: Even reduction in pipeline thickness; varying axial length (0.15~1.5 m) and depth (0.5~5 mm).Pitting: Micrometer-scale localized corrosion (pitting); varying radius (0.2~1 mm) and depth (0.5~5 mm).

By performing these simulations, a deeper understanding can be gained of the different types of defects in pipelines and their effects on pipeline integrity. Additionally, the data collected from these simulations can be used to build a comprehensive training dataset for further defect classification and prediction work. This dataset can then be leveraged to develop more accurate and reliable diagnostic tools, ultimately helping to improve pipeline safety and maintenance practices.

Here, Figure 6 illustrates a quasi-distributed sensing scheme consisting of five fiber sensing regions arranged in 1200 (time) × 5 (sensor) matrices. Comparison of the acoustic waveform from these sensors includes information about attenuation and reflection shown in Figure 6. Pitting corrosion shows only a relatively weak signal due to small dimensions of the assumed defect, which can be a significant practical challenge in pipeline health detection. Of the five events, the reflection signal generated by welding is the most pronounced. In contrast, generalized and localized corrosion produces a range of reflected waves due to structural discontinuities. Based on the results of the analysis, we observed that the presence of clamps in the blue region of the pipeline observed led to a reduction in the amplitude of the resulting signal. This reduction can be attributed to the elastic support provided by the clamps, which helps to suppress the reflection or distortion of the wave. However, some weak echo signals were also observed in the resulting signal, indicating that there may still be some scattering or reflection of the wave occurring in the clamped region. In the following sections of the paper, the authors will describe specific methods and processes used to apply neural networks and training techniques to identify pipe events through analysis of guided wave interactions. This study focuses primarily on the extraction of simulated Distributed Acoustic Sensing (DAS) system data, with the intent of developing a data-driven method for pipeline event classification. The preprocessing of training data is an essential step in this process to ensure its suitability for the neural network model. The emphasis is not on CNN model structure at this stage. Instead, we aim to showcase how supervised learning-based neural networks can effectively identify and classify pipeline events. This study serves as a proof-of-concept for the proposed method, paving the way for its further development and refinement in future work.

### 5.2. Data Pre-Processing

The accuracy of fiber optic sensors can be negatively impacted by noise, which can arise from various factors such as light source, coupling efficiency, and signal processing amongst other sources [36]. To mitigate the effects of noise, it is important to identify and analyze different sources of noise and take appropriate measures to minimize their impact. While a detailed analysis of each contributing factor can be valuable, it may also be possible in some cases to define a reasonable noise level based on experimental data. This can be achieved through experimentation and validation of the chosen noise level to ensure accurate and reliable results. In the present study, the challenge of real-world noise in data collected from an acoustic fiber sensor experiment was addressed, considering various random processes such as source noise, detector noise, and background noise. Current fully distributed DAS interrogation systems are not capable of operating at frequencies of 30–50 kHz. Thus, in the current study we use a representative example of the magnitude of experimental noise derived from a previously demonstrated experimental setup having sufficient frequency bandwidth for ultrasonic acoustic guided wave monitoring, and its resulting signal with a 32 kHz excitation as illustrated in Figure 7a,b [10]. In this case, a quasi-distributed multimode interferometric structure is assumed. In future studies, deeper investigation of noise levels will be conducted for fully and various quasi-distributed acoustic fiber optic sensing schemes.

The sensor structure assumed here is composed of a multimode interferometer using a DFB laser with an output power of 45 mW as a laser source and a single wavelength laser output split into N paths using a 1 × N fiber coupler. Validation of the sensor signal derived experimentally in past work on a 50-foot length of pipeline theoretically is also presented in Figure 7c for comparison. We use the experimental data to extract a rough estimate of noise from the sensor signal resulting in an estimated value of approximately 9.63 dB. Based upon this estimate, the basic noise model of additive white Gaussian noise (AWGN) was used in training data and compared to data without noise introduced. The noise in decibels (*dB*) is defined as a logarithmic representation of signal-to-noise ratio:(8)$$SNR_{dB}=10log_{10}\left(\frac{P_{signal}}{P_{noise}}\right)$$

Here, *P_signal_* is the experimental signal after passband filtering the raw data set as per our prior publications [10,37]. The noise signal, *P_noise_*, can be extracted from the received signal by applying wavelet transformation to isolate different frequency components, followed by bandpass filtering to further remove unwanted frequency components outside the specified range. This processed signal can then be used to estimate noise and calculate signal-to-noise ratio. Figure 8 shows how the simulated signal is affected with application of background noise, in this case noise extracted from recently completed experimental measurement of guided acoustic waves on a pipeline by a quasi-distributed sensor.

After generating the simulated data, we construct a training data matrix by considering each time-domain signal of a fiber segment along a specific spatial resolution limit as a separate row. This spatial resolution limit, also referred to as the minimum resolvable distance or the spatial sampling interval, functions as a low-pass filter for the data that can be acquired. This process can be generalized for both fully distributed and quasi-distributed sensing systems. In the data matrix, the spatial domain is represented by the vertical column, while the time domain is represented by the horizontal row. Figure 9 illustrates the temporal-spatial data matrix for a healthy pipeline. Figure 9a,c display the signal matrix from a fully distributed fiber sensor with and without noise interference (SNR = 9.63 dB), respectively. For Figure 9b,d, the received data from the quasi-distributed sensing system has the same matrix size as the fully distributed sensor signal contour. Along the pipeline, there are 12 sensor segments, corresponding to 12 signal channels in the signal matrix plot. The invalid area in the sensor fiber, depicted as a blue segment in Figure 9, is filled with zero values in the dataset.

The data for each pipeline segment is represented as a matrix with a size of 96-inch in the spatial domain and 1.2 ms (1200-time steps) in the temporal domain. These matrices are converted into greyscale images with a size of 46 × 1200 pixels before being fed into the Convolutional Neural Network (CNN) in the case of both training and test data. Resultant grayscale images represent different events, such as welds, clamps, and corrosion, and are shown in Figure 10.

## 6. Event Recognition

### 6.1. Comparison of Common CNNs

Convolutional neural networks (CNNs) have gained significant popularity in recent years due to their remarkable effectiveness in processing and classifying signals, such as time-series data from sensors or audio signals. This can be attributed to the unique ability of CNNs to capture local patterns and time dependencies in the input data while maintaining robustness to changes in the timing and amplitude of the signal. As a result, they are particularly well-suited for processing complex and noisy signals that are frequently encountered in real-world applications.

In our research, our central goal is to evaluate the performance of training data (both noise-free and noisy data from Das and qDas systems) in differentiating six distinct categories of pipeline signals using Convolutional Neural Network (CNN) models. Given the investigative nature of this study, our inquiries are based on a relatively small dataset. It is important to acknowledge that while the size of the dataset and the choice of the model play crucial roles in shaping our research, these elements are defined within the limitations of this initial effort. As we progress with this framework, we anticipate refining our model selection process and utilizing a more expansive dataset.

The CNN model employed in our study is configured with a single layer, a kernel size of 10, and the ‘ReLu’ activation function, as depicted in Figure 11. This architecture was chosen to leverage the ‘ReLu’ activation function’s efficiency in counteracting the vanishing gradient problem, a common challenge in more profound networks. Our single-layer model strikes an optimal balance between simplicity and effectiveness, offering computational efficiency while adeptly identifying critical features within the input signals.

Our dataset comprises 150 samples, each characterized by dimensions of 46 (space) × 1200 (time points), free from noise interference. To accommodate the dataset’s limited size, we implemented a stratified sampling strategy when partitioning the data into training and test sets, thereby preserving the distribution of the six distinct types of pipeline signals. We also undertook a systematic approach to data pre-processing, which involved normalizing input features and eliminating potential outliers and inconsistencies.

Despite the dataset’s limitations, we executed the training process for our CNN model over 1000 iterations, allowing us to balance computational feasibility with robust performance estimation. We further reinforced the reliability of our results through k-fold cross-validation, enhancing their generalizability. This methodology served to mitigate overfitting risks and provide a more accurate estimation of model performance. In the following section, we will delve into a detailed analysis of the prediction accuracy performance of our chosen CNN model.

### 6.2. The Impacts of Classification Accuracy Due to Sensing System to the Robustness of Data Classification

In this section, we compare the classification accuracy of two different acoustic sensing systems: a fully distributed acoustic sensing (DAS) system and a quasi-distributed acoustic sensing (qDAS) system. The DAS system provides continuous measurements along the entire length of the pipeline, while the qDAS system uses a limited number of sensors placed at discrete locations along the pipeline. We evaluate accuracy of these systems in classifying the condition of the pipeline based on six different defect types. It is noted that fully distributed sensing capability is currently limited for ultrasonic guided wave acoustic monitoring due to both the large gauge length in current standard commercial systems (~1 m) and the limited acoustic frequency bandwidth (~10 kHz). Nevertheless, we include hypothetical fully distributed sensing schemes for completeness.

Figure 12 and Table 4 underscore the classification prowess of our Convolutional Neural Network (CNN) model, particularly when it comes to identifying a range of pipeline features. This performance remains consistent across both fully distributed and quasi-distributed acoustic sensing systems. The categories of welding, localized corrosion, and general corrosion saw particularly strong results, with our CNN model demonstrating high accuracy in both systems. This is a significant finding, as these types of defects—especially localized and general corrosion—can severely impact the structural integrity and lifespan of the pipeline system. Therefore, the ability of our classifier to correctly identify these types of corrosion is crucial. Nevertheless, despite these promising results, our research into enhancing the classification of these corrosion types remains ongoing, and we continue to explore ways to further improve the model’s performance in these areas.

Additionally, it is important to consider advanced performance measures such as the implications of false negatives when analyzing the performance of the CNN model. A higher false negative rate (FNR) implies that the model is more likely to miss actual instances of a particular defect category. Given the potential for significant impacts of false negatives in terms of corrosion detection in terms of health and human safety as well as economic costs, this measure is of unique importance in pipeline structural health monitoring frameworks. In the given table, the FNR values are therefore also presented. Additional performance measures can also be developed and considered in future work. Based upon the performance measures presented, a high degree of accuracy and a low FNR is achieved for both localized and generalized corrosion defects and welds, with lower accuracy observed for pitting defects, particularly for the quasi-distributed scheme.

Results of classification performance benchmarking comparisons between fully and distributed sensing points to the importance of considering the spatial resolution and sensitivity of the sensing systems when dealing with subtle features such as pitting corrosion. Ensuring the detection system’s ability to accurately capture these features will be crucial to achieving classification accuracy for these types of defects.

The categories of ‘clamp’ and ‘no defect’ also display high classification accuracy in both systems, with only a slight drop in precision for the ‘no defect’ category in the quasi-distributed system and a slight increase in FNR for the ‘clamp’ category. Despite these minor deviations, the model’s overall performance remains robust, demonstrating its ability to effectively differentiate between various types of pipeline features.

### 6.3. Analysis of Classification Performance with Noise Effect

In this scenario we introduce noise and again present two confusion matrices and classification reports for a fully distributed sensor system and a quasi-distributed sensor system (Figure 13). Sensor systems are affected by Gaussian noise with a signal-to-noise ratio of 9.63 dB. The training dataset for the CNN model is shown in Figure 9c,d.

The confusion matrices and classification reports provide insights into the performance of fully distributed and quasi-distributed sensor systems in classifying six distinct types of defects in a pipeline condition monitoring application. When comparing the two confusion matrices, it is apparent that the fully distributed system boasts a higher overall accuracy of 98% as opposed to the 96% accuracy of the quasi-distributed system. This difference in accuracy is further reflected in the precision and recall values for most defect categories, with the fully distributed system typically outperforming its counterpart. For instance, in Table 5, the fully distributed system exhibits superior recall and precision for welding, localized corrosion, and pitting corrosion, signifying its ability to correctly classify all instances within these categories. Conversely, the quasi-distributed system displays lower recall values for these categories but demonstrates improved performance for localized corrosion, general corrosion, and clamps. The enhanced performance of the fully distributed system in the pitting corrosion category is particularly notable.

However, the quasi-distributed system’s better performance in classifying localized corrosion and general corrosion may be attributed to the impact of noise, which could affect the fully distributed system more significantly than the quasi-distributed one in certain assumed sensor network configurations. This observation underlines the importance of understanding the effects of noise and other environmental factors on sensor system performance when dealing with real-world applications.

Detailed investigations of various sensor network configurations and their individual effects on model performance are crucial aspects of optimizing the sensor system’s performance for specific sensing objectives. Factors such as noise resistance, sensing coverage, and data processing capabilities should be considered when selecting the most suitable sensor network for the task at hand. In conclusion, the analysis of the confusion matrices highlights the importance of selecting the appropriate sensor network configuration for accurate defect classification in pipeline condition monitoring applications. While the fully distributed system appears to outperform the quasi-distributed system in most categories, understanding the unique strengths and weaknesses of each system is crucial to ensure optimal performance in real-world scenarios.

### 6.4. Analysis of Classification Performance with Noise and Varying Quasi-Distributed Sensing

In this study, we have explored the performance of quasi-distributed acoustic sensing systems compared to fully distributed systems. Quasi-distributed systems utilize a limited number of strategically placed sensors to provide a balance between performance and resource utilization. We conducted a comprehensive analysis of the prediction accuracy of quasi-distributed systems at different sensor configurations, investigating the impact of sensor segments on prediction accuracy.

To investigate the machine learning classifier based on ultrasonic guided acoustic wave and fiber optic sensor fusion in more detail, we consider the possibility of varying positions and number of quasi-distributed sensors, which can be adjusted as shown in Figure 14 as red segments of 1, 4, 7, 10, 13, …, 40, 43, 46 (Continuous DAS) at an assumed noise level (SNR = 9.63 dB). The configuration shows a fiber sensor that completely covers the pipe with measurement units evenly distributed along its length. 

To evaluate their performance, a series of tests was conducted, and the results are presented in Figure 15 and described in more detail below. In addition to the total sensor segment numbers, we also explored the effect of sparse sampling on the acoustic sensing system’s ability to withstand variations in sensor positioning. To evaluate the influence of sensor positioning at varying signal-to-noise ratios, we uniformly selected sensor segments for a specific sensor count and carried out the selection procedure 10,000 times at a consistent noise level. We examined 16 distinct sensor numbers, ranging from 1 to 46, with sparse sampling playing a role in this selection. By creating boxplots of the dataset, we could visually appraise the distribution of prediction accuracy across various SNR and sensor numbers. These figures offer insight into the performance fluctuations for each configuration and the system’s ability to withstand alterations in sensor placement, with sparse sampling potentially contributing to the observed variations.

Figure 15 shows the impact of random down spatial sampling of received signals on classification prediction using a CNN under various levels of background noise. Our results demonstrate that as the level of background noise increases, prediction accuracy becomes less stable, with an increased variance and more outliers. Furthermore, as the sampling size increases, the diversity of random sampling decreases, thereby improving the robustness of multiple predictions. However, increasing the spatial size of the sampling from 16 to 46 sensor segments did not significantly improve prediction accuracy, and in some cases, a specific small subset of the training set achieved higher accuracy than a fully sampled (46 sensor segments) training set.

Our findings reveal a trade-off between the number of sensor segments and prediction accuracy in the context of sparse sampling. However, this improvement becomes less significant as the number of sensors rises. Engineers can effectively tailor the design of quasi-distributed acoustic sensing systems for diverse applications by considering the signal-to-noise ratios and the number of sensors to achieve an optimum between predictive accuracy and reduced system intricacy and resource requirements using sparse sampling.

Furthermore, our findings align with the work of Jingwen Hu (2015) [38], who suggested that scene classification is composed of various scenes, ranging from simple to complex. Under hardware limitations, a random sampling strategy is recommended due to its robustness, good performance, and lower spatial complexity. However, random sampling may not meet specific requirements for sampling and lacks interpretability.

Down-sampling or sparse-sampling has been investigated in several studies, with consistent and interesting results. For instance, Cohen et al. (2018) [39] demonstrated the advantages of spatial and temporal down-sampling in event-based visual classification, while Kang et al. (2020) [40] examined the effects of uniform down-sampling in a deep CNN-based ground vibration monitoring scheme for MEMS sensed data. Similarly, Naagome et al. (2020) [41] showed that down-sampling increased the accuracy of RNNs in decoding gait from EEG data. These studies emphasize the importance of down-sampling as a preprocessing technique for improving the efficiency and accuracy of neural networks in various applications.

Our research can be extended in future work to optimize sparse sensor placement for classification and transition from sparse to dense sampling in compressed measurements [42]. By leveraging these insights, researchers and engineers can develop more cost-effective and efficient systems for a range of applications, while optimizing resource allocation.

## 7. Conclusions

This proof-of-concept study explores the potential of machine learning to accelerate the classification of damage size and orientation in guided wave-based damage detection methods by combining ultrasound acoustic guided wave-based non-destructive evaluation methods with distributed/quasi-distributed fiber optic acoustic sensing. The proposed learning framework provides an efficient workflow and identifies potential areas of improvement to develop a robust and experimentally validated framework. The investigation explores the impact of noise interference, mixed data types, and various features and corrosion defects on the efficacy of the proposed method.

The effect of noise on the prediction accuracy and sensor type (fully vs. quasi) of the sensor system was also investigated and shows significant differences in performance depending upon the specific assumptions. Our results show that fully distributed acoustic sensing systems exhibit higher overall accuracy and precision compared to quasi-distributed systems. In addition, the presence of experimentally relevant noise levels adversely impacted the overall accuracy and precision but did not preclude a high level of performance. The performance gap narrows specifically for detecting specific types of defects, with the quasi-distributed systems investigated being particularly effective for localized corrosion and general corrosion even in the presence of experimentally relevant noise.

Regarding the sensor location and its effect on accuracy, our results show that the performance of quasi-distributed acoustic sensing systems can be significantly affected by the sensor location and number. In general, the first few additional sensor elements improve classification framework accuracy significantly, and eventually the additional improvement becomes limited or even negligible for sufficiently large numbers of sensors approaching a fully distributed sensing configuration. Sparse sampling strategies can be effectively utilized to balance prediction accuracy with reduced system complexity and resource requirements. Understanding trade-offs and optimizing sensor networks is an area in which additional future work can be pursued, as understanding the impact of noise and sensor placement on the performance of fully distributed and quasi-distributed systems is critical to optimizing sensor network configurations and achieving reliable, robust, and accurate data classification in pipeline monitoring applications.

## Figures and Tables

**Figure 1 sensors-23-05410-f001:**
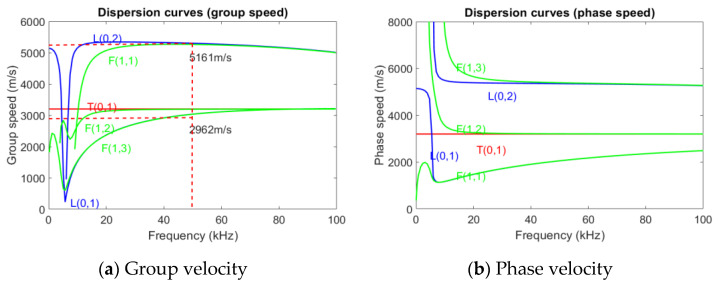
Dispersion curves for a 0.5 inches thickness and 12 inches outside diameter pipeline model (**a**) group velocity; and (**b**) phase velocity.

**Figure 3 sensors-23-05410-f003:**
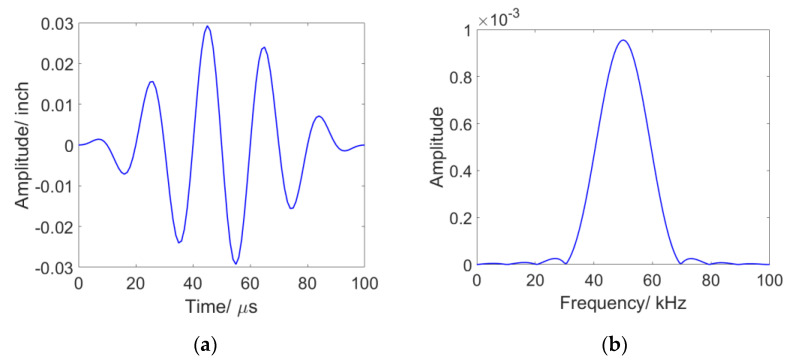
The excitation signal assumed in the simulation; (**a**) the waveform in the time domain; (**b**) the corresponding wave in the frequency domain.

**Figure 4 sensors-23-05410-f004:**
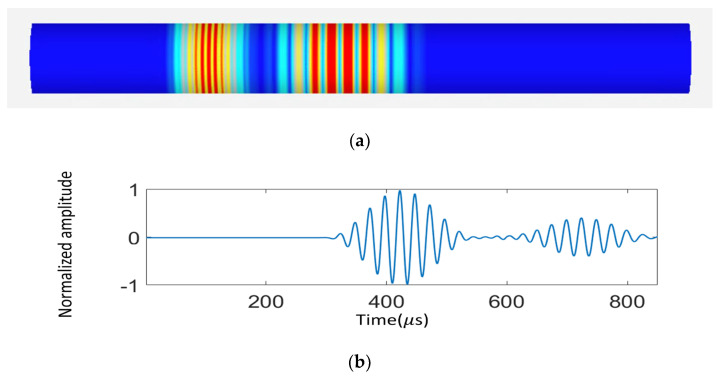
Comparison of the literature result [28] and the excitation signal dispersion from the proposed simulation work. (**a**) Excitation signal dispersion contour from the proposed simulation, (**b**) Dispersive time domain signal from the proposed simulation, (**c**) Comparison of the dispersive signals from the literature [31] and the proposed simulation. (Excitation: (**a**,**b**) A five-cycle Hann-windowed 50 kHz signal; (**c**) A ten-cycle Hann-windowed 40 kHz signal).

**Figure 5 sensors-23-05410-f005:**
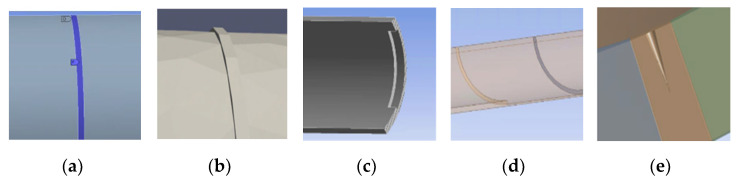
Illustration of pipe structure type classification: (**a**) Clamp (~inch); (**b**) Welding (~inch); (**c**) Localized Corrosion (~inch); (**d**) General Corrosion (~feet); (**e**) Pitting Corrosion (radius: ~200 µm).

**Figure 6 sensors-23-05410-f006:**
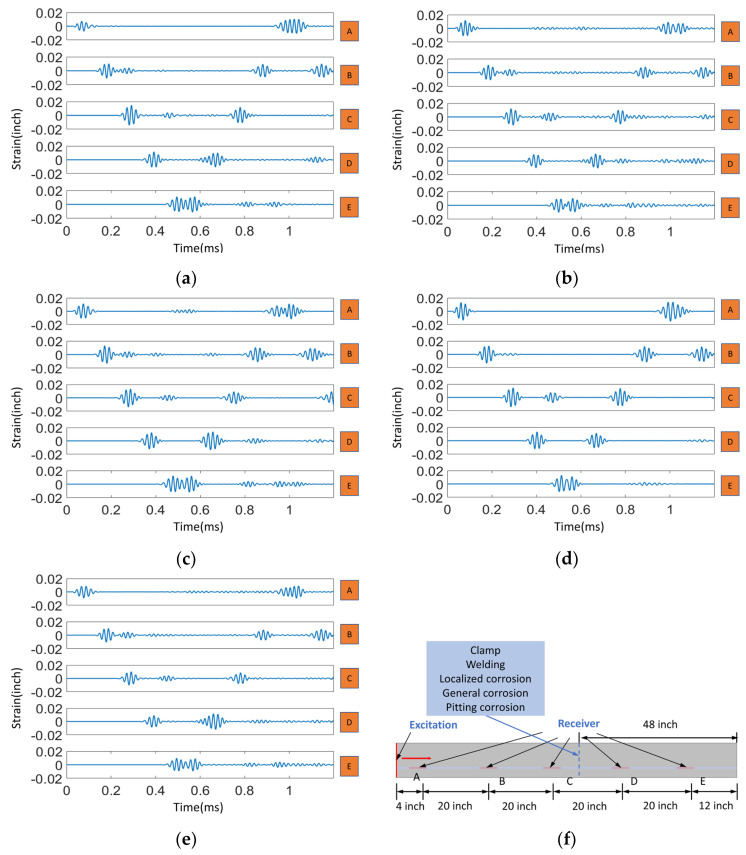
The time-domain acoustic waveforms from 5 sensors (A–E) by the excitation and the typical reflected signals of the five types of damage: (**a**) echoes of clamp; (**b**) echoes of welding; (**c**) echoes of localized corrosion; (**d**) echoes of general corrosion; (**e**) echoes of pitting corrosion; and (**f**) the schematic diagram of fiber optic acoustic sensors positions for pipeline structure monitoring.

**Figure 7 sensors-23-05410-f007:**
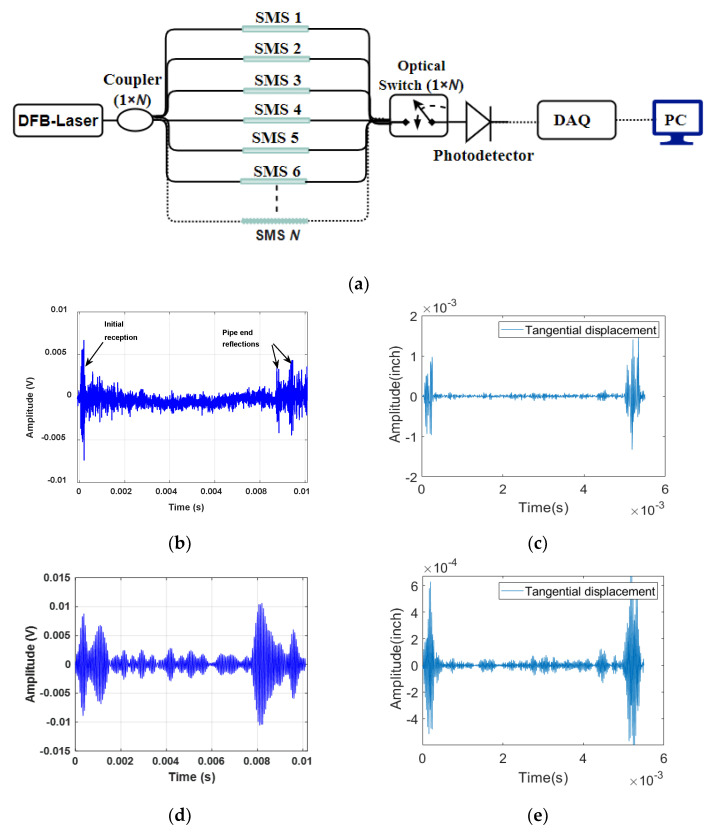
Comparative Analysis of Experimental and Simulated Results for a 32 kHz Quasi-Distributed SMS Fiber Sensor: (**a**) Multiplexed Interrogator Schematic; (**b**) Received Signal with 32 kHz Excitation; (**c**) Simulated 32 kHz, 5-Cycle Sinusoidal Signal for Validation at Same Location as Experiment; (**d**) Time-Domain Signal after Passband Filtering (26–36 kHz); (**e**) Filtered Simulated 32 kHz, 5-Cycle Sinusoidal Signal (26–36 kHz).

**Figure 8 sensors-23-05410-f008:**
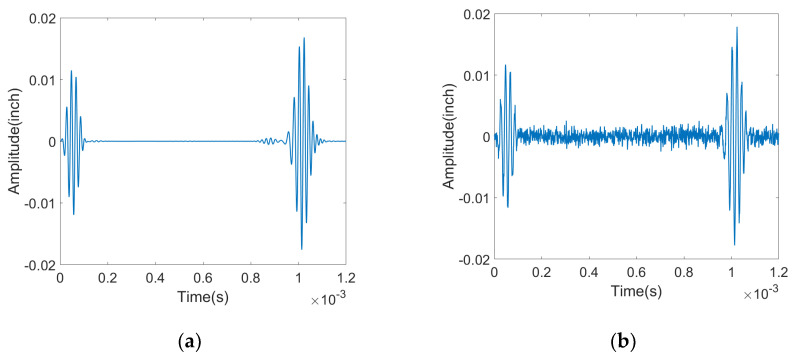
(**a**) Simulated Pure Signal—Original signal without any noise interference, (**b**) Signal with 9.63 dB Noise Added—the pure signal is altered by introducing a 9.63 dB noise component to simulate real-world conditions.

**Figure 9 sensors-23-05410-f009:**
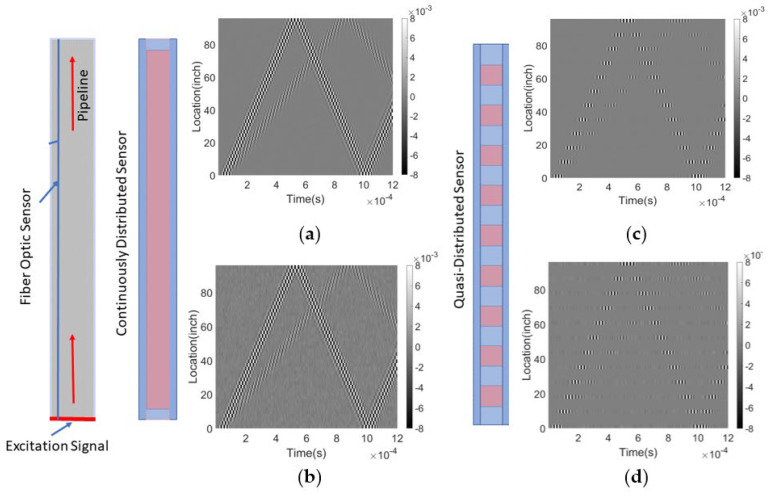
Resulting signal for a healthy pipeline based on distributed sensing system (**a**) without noise and (**b**) with Gaussian noise (SNR = 9.63 dB); and the resulting signal based on quasi-distributed sensing system (12 signal channels) (**c**) without noise and (**d**) with Gaussian noise (SNR = 9.63 dB). The resulting signal is shown as a 2D time-space plot, where the x-axis represents time, and the y-axis represents the length of the pipeline.

**Figure 10 sensors-23-05410-f010:**
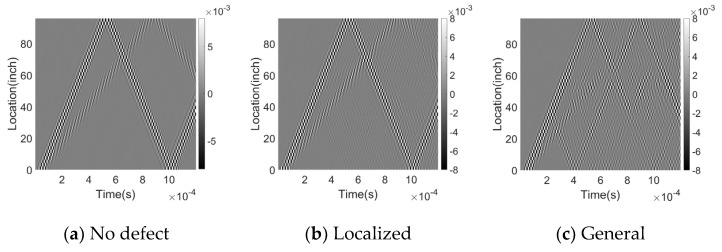
Time-space plots for a pipeline with (**a**) no defect, (**b**) localized corrosion, (**c**) general corrosion, (**d**) pitting corrosion, (**e**) welding, and (**f**) clamp. Each plot shows the resulting signal from a different type of defect or feature. The x-axis represents time, and the y-axis represents the spatial location along the pipeline. The color scale indicates the amplitude of the signal.

**Figure 11 sensors-23-05410-f011:**
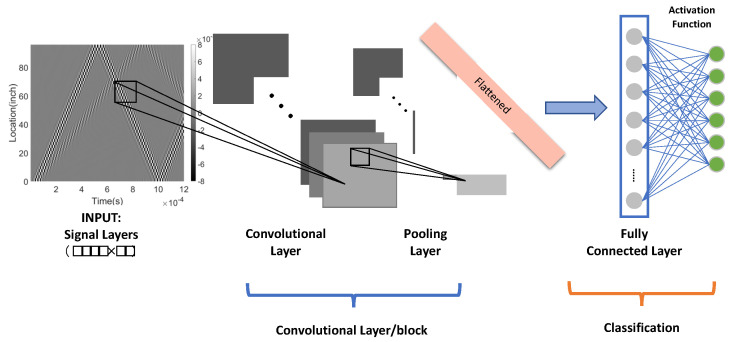
One-Layer CNN for Signal Classification: Input data is a 2D tensor of size (batch size, 1200, 46).

**Figure 12 sensors-23-05410-f012:**
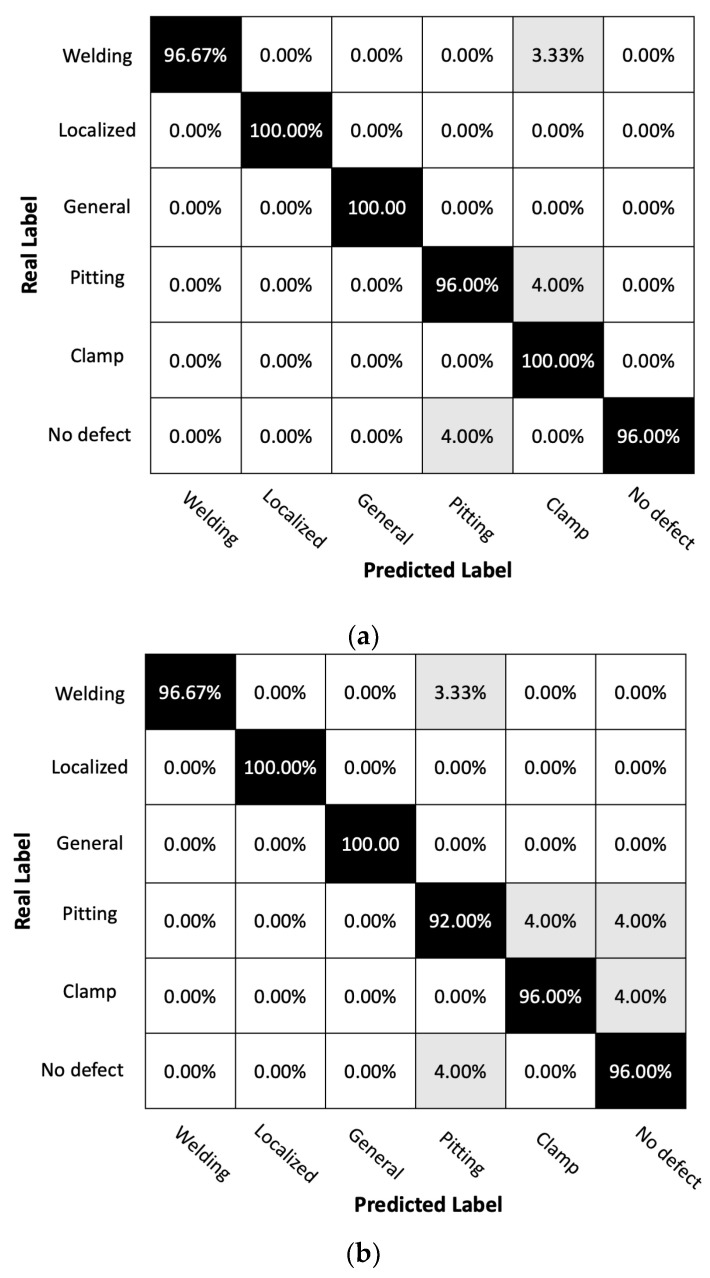
Matrix for pipeline condition classification using (**a**) fully distributed acoustic sensing (DAS) system and (**b**) quasi-distributed acoustic sensing (qDAS) system with 12 sensor segments. The confusion matrix shows the number of true positives, true negatives, false positives, and false negatives for each pipeline condition class. The classification accuracy for fully DAS system is 99% and for quasi-DAS system is 96%. The results suggest that the fully DAS system achieved higher classification accuracy compared to the qDAS system, particularly in distinguishing between different types of corrosion.

**Figure 13 sensors-23-05410-f013:**
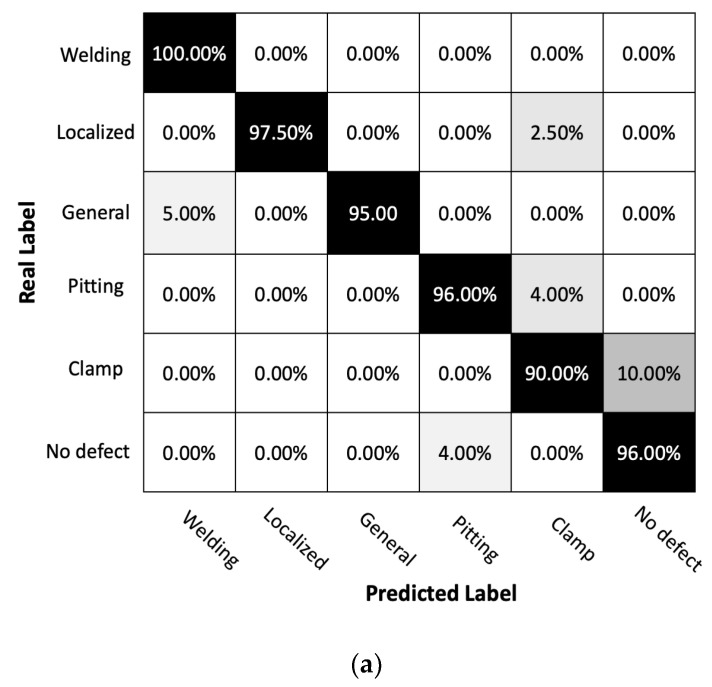
Comparison of Confusion Matrices for (**a**) Fully Distributed and (**b**) Quasi-Distributed Sensor Systems with Gaussian Noise and SNR = 9.63 dB.

**Figure 14 sensors-23-05410-f014:**
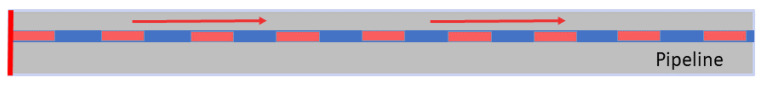
The sensor system is configured as a fiber sensor that covers the pipe, with the measurement units evenly distributed along its length.

**Figure 15 sensors-23-05410-f015:**
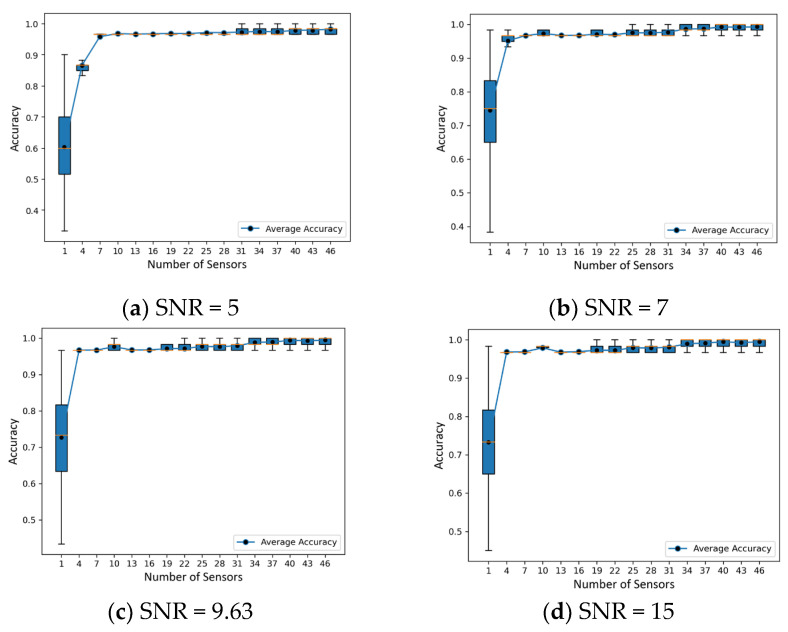
The prediction accuracy changes with different sensor numbers under different signal-noise ratio.

**Table 1 sensors-23-05410-t001:** Dimensional parameters of the steel pipeline model.

Length (m)	Outside Diameter (m)	Wall Thickness (m)
2.44	0.3	0.013

**Table 2 sensors-23-05410-t002:** Material property parameters of the steel pipeline model.

Material	Density ($kg/m^{3}$)	Young’s Modules (Pa)	Poisson Ratio (μ)
	7850	$21\times 10^{10}$	0.32

**Table 3 sensors-23-05410-t003:** Dimensional parameters of pipeline feature types.

Pipe Feature	Clamp	Welding	Localized Corrosion	General Corrosion	Pitting
Variable	Axial length: (0.5~5 mm) Stiffness factor: ($5\times 10^{6}$~$6.5\times 10^{6}$ N/m)	Axial length: (0.5~5 mm) height: (0.5~5 mm)	Axial length: (25.4~152 mm), Depth: (0.5~5 mm)	Axial length: (0.15~1.5 m) Depth: (0.15~5 mm)	Radius: (0.2~1 mm) Depth: (0.5~5 mm)
Case Number ^1^	20	30	40	40	25
Description	Elastic support loaded by clamps (blue region);	Discontinuity and material property changes between the pipeline and the welded portion	A rectangular notch on the inner surface of the pipeline, indicating presence of corrosion in a specific area.	Even reduction in pipeline thickness	Micrometer-scale localized corrosion of a specific type

^1^ Multiple simulation runs are conducted of defects in batches, where each group’s variables—such as the defect width or material properties of the clamp part—are randomly altered.

**Table 4 sensors-23-05410-t004:** Comparison of Classification Performance Metrics for Fully Distributed and Quasi-Distributed Sensor Systems without Noise.

	Fully Distributed			Quasi-Distributed		
Classification report	Precision	Recall	FNR	Precision	Recall	FNR
Welding	100.00%	96.67%	3.33%	100.00%	96.67%	3.33%
Localized corrosion	100.00%	100.00%	0.00%	100.00%	100.00%	0.00%
General corrosion	100.00%	100.00%	0.00%	100.00%	100.00%	0.00%
Pitting corrosion	96.00%	96.00%	4.00%	96.67%	92.00%	8.00%
Clamp	94.34%	100.00%	0.00%	96.00%	96.00%	4.00%
No defect	96.00%	96.00%	4.00%	92.31%	96.00%	4.00%

**Table 5 sensors-23-05410-t005:** Comparison of Classification Performance Metrics for Fully Distributed and Quasi-Distributed Sensor Systems with Gaussian Noise, SNR = 9.63 dB.

	Fully Distributed			Quasi-Distributed		
Classification report	Precision	Recall	FNR	Precision	Recall	FNR
Welding	95.24%	100.00%	0.00%	100.00%	93.33%	6.67%
Localized corrosion	100.00%	97.5%	2.50%	97.50%	97.50%	2.50%
General corrosion	100.00%	100.00%	0.00%	93.94%	97.50%	2.50%
Pitting corrosion	96.00%	100.00%	0.00%	90.00%	90.00%	10.00%
Clamp	94.34%	90.00%	10.00%	96.00%	96.00%	4.00%
No defect	100.00%	96.00%	4.00%	90.00%	90.00%	10.00%

## Data Availability

The data presented in this study are available within the article.

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
