# Peer review of "Quasi-Distributed Fiber Sensor-Based Approach for Pipeline Health Monitoring: Generating and Analyzing Physics-Based Simulation Datasets for Classification"

_sensors, 2023, doi:10.3390/s23125410_

Round 1

Reviewer 1 Report

The authors proposed real-time structural change detection using CNNs employing acoustic fiber sensors. The reviewer has the following concerns & questions:

1.  The abstract is poorly written. Please articulate it in a way that conveys the structure of the paper concisely. 

2.  Divide the subsections into three major categories: 1. Sensing system, 2. Data Generation 3. Data modelling or damage classifications.

3.  The paper is very generic and does not capture the machine learning aspect.  For example, there is no section on hyperparameter optimization and the range of hyperparameters used and why? 

4. The performance measures of the ML model are very generic, in case of pipeline damage where human life is at stake, don't the authors think, having a "false negative" as one of the performance measures would be more critical than precision or F1 score? If your model is not capable of detecting one kind of damage, and it leads to catastrophic failure, would you be willing to use this model?

5. If you have 100% accuracy, either you are overfitting or you have a small dataset. Please provide clarification. 

6. It is common knowledge that Rectified Linear Unit (ReLu) is because the vanishing gradient problem is completely removed in the activation function which makes this activation function more advanced as compared to sigmoid or tanH. I think training CNN with what has already been established is not acceptable. 

For these reasons, I am rejecting this paper

ok

Reviewer 2 Report

This study proposes an effective framework for detecting mechanical damage in pipelines, with a valuable workflow for damage detection and identification. The use of physics-informed datasets enhances the accuracy and reliability of the model, and the investigation of the impact of sensing systems and noise on classification performance adds to its robustness. The study is a beneficial step forward in pipeline maintenance and enhances the safety and security of pipeline operations.

1. To ensure the accuracy of the calculation and consider the influence of calculation, in this paper, the general mesh size should be smaller than 1/20 of the minimum wavelength along the pipeline body. But this may significantly increase computational complexity, and neural networks can also have a significant burden. Will this affect recognition efficiency?

2. inch does not seem to belong to the SI unit system, please replace it with mm.

3. It seems that many artifacts can be seen in Figure 9. Is there any way to eliminate them?

4. The Introduction appears to lack sufficient literature, including recent research.

Round 2

Reviewer 1 Report

The authors have improved the manuscript considerably.  

English is ok.

Reviewer 2 Report

The paper is well revised and can be accepted for publication.